# Finding shortest and nearly shortest path nodes in large substantially incomplete networks by hyperbolic mapping

Maksim Kitsak [1,2] ✉, Alexander Ganin [3,4], Ahmed Elmokashfi [5], Hongzhu Cui [6,7], Daniel A. Eisenberg [8], David L. Alderson [8], Dmitry Korkin [6,9,10] & Igor Linkov [11] ✉

Dynamic processes on networks, be it information transfer in the Internet, contagious spreading in a social network, or neural signaling, take place along shortest or nearly shortest paths. Computing shortest paths is a straightforward task when the network of interest is fully known, and there are a plethora of computational algorithms for this purpose. Unfortunately, our maps of most large networks are substantially incomplete due to either the highly dynamic nature of networks, or high cost of network measurements, or both, rendering traditional path finding methods inefficient. We find that shortest paths in large real networks, such as the network of protein-protein interactions and the Internet at the autonomous system level, are not random but are organized according to latent-geometric rules. If nodes of these networks are mapped to points in latent hyperbolic spaces, shortest paths in them align along geodesic curves connecting endpoint nodes. We find that this alignment is sufficiently strong to allow for the identification of shortest path nodes even in the case of substantially incomplete networks, where numbers of missing links exceed those of observable links. We demonstrate the utility of latent-geometric path finding in problems of cellular pathway reconstruction and communication security.

Being the tallest building in the Western Hemisphere, the One World Trade Center (OWTC) is easily observable from virtually any point of the lower Manhattan island. Tourists can find their way to the building without a map as long as it stays in their line of sight. Another tourist attraction is the Peace Maze in Northern Ireland. The maze's exit is placed in the center and can also be spotted from anywhere within the maze. Nevertheless, finding the exit path from the maze is not straightforward. From the graph theory perspective, both the road system of Manhattan and the Peace Maze are graphs or networks, and both problems reduce to finding the shortest path connecting the origin with the destination. What makes New York City's most densely populated borough navigable is the geometric grid-like structure of its

[1]Faculty of Electrical Engineering, Mathematics and Computer Science, Delft University of Technology, 2600 GA Delft, The Netherlands. [2]Network Science Institute, Northeastern University, 177 Huntington avenue, Boston, MA 022115, USA. [3]University of Virginia, Department of Systems and Information Engineering, Charlottesville, VA 22904, USA. [4]U.S. Army Engineer Research and Development Center, Contractor, Concord, MA 01742, USA. [5]Simula Metropolitan Center for Digital Engineering, Oslo, Norway. [6]Bioinformatics and Computational Biology Program, Worcester Polytechnic Institute, Worcester, MA 01609, USA. [7]Institute for Genomic Medicine, Columbia University Medical Center, New York, New York, USA. [8]Department of Operations Research, Naval Postgraduate School, Monterey, CA 93943, USA. [9]Computer Science Department, Worcester Polytechnic Institute, Worcester, MA 01609, USA. [10]Data Science Program, Worcester Polytechnic Institute, Worcester, MA 01609, USA. [11]U.S. Army Engineer Research and Development Center, Environmental Laboratory, Concord, MA 01742, USA. ✉e-mail: maksim.kitsak@gmail.com; igor.linkov@usace.mil

intersections. The line of sight to the OWTC is nothing else but the geodesic curve connecting the tourist's current location to its destination point, and the tourist may find her way to the OWTC by taking the streets with minimal deviation from this geodesic.

Different from the Manhattan road network, links in many real networks, such as the Internet, social networks, and networks of molecular interactions, are not determined by physical proximities of their nodes. On the contrary, these networks are characterized by effective geometries, which are often referred to as latent or hidden. Nodes in these networks can be mapped to points in latent spaces by an optimization procedure, often called network embedding, such that in the resulting map, network links are likely to connect nodes separated by small distances in the latent space[1,2]. Recent works indicate that common topological properties of real networks, such as the hierarchical organization, the heterogeneity in the number of connections per node, strong clustering coefficient, and self-similarity[3], are best mapped into latent spaces, which are hyperbolic rather than *Euclidean*. Notable examples of real networks with effective hyperbolic geometries are the PPI networks[4], the Internet[5], and social networks[6,7]. At the same time, there is no consensus on the extent to which shortest paths in these networks align along geodesic curves. While an earlier work demonstrated that hyperbolic geometry could be used to find Internet routing paths[5], more recent work finds that the topologically shortest paths are statistically different from geometrically shortest paths[8].

In this work, we find that shortest paths in real networks—the Internet at the Autonomous System level, the human similarity-based PPI network, and the Pretty-Good-Privacy (PGP) web of trust—display geometric localization in their hyperbolic representations. Nodes constituting shortest paths in these networks can be directly identified by their proximity to the geodesic connecting the shortest path endpoints: the closer the node to the geodesic the higher the likelihood it belongs to a shortest path. We establish that distance to geodesic is sufficiently accurate in finding shortest path nodes in situations when large fractions of network links are missing in the network data. We demonstrate that geometric localization of shortest paths can be used to validate routing paths in the Internet and reconstruct cellular pathways.

## Results

### Distance to geodesic and path-finding accuracy

To demonstrate the main finding, we first visualize two shortest paths of the AS-level Internet in its 2-dimensional hyperbolic representation. In the AS-level Internet, nodes are Autonomous Systems (ASes), and connections between them are contractual agreements governing data flows between ASes, Section SV and Supplementary Data 1.

Figure 1a demonstrates that nodes comprising the two shortest paths are not random but tend to lie in the geometric vicinity of corresponding hyperbolic geodesics connecting the endpoints of the two shortest paths. To quantify the observed alignment, we measured distances from the shortest path nodes to the geodesic curve, Fig. 1b. To do so, we employed an approximation for a distance from point $C$ to the hyperbolic geodesic curve $\gamma(A, B)$ connecting points $A$ and $B$ in hyperbolic disk $\mathbb{H}^2$:

$$d(C, \gamma(A,B)) = \frac{1}{2}\left[d(A, C) + d(B, C) - d(A, B)\right] + \ln 2, \quad (1)$$

where $d(X, Y)$ is the distance between points $X$ and $Y$ in $\mathbb{H}^2$, see Methods.

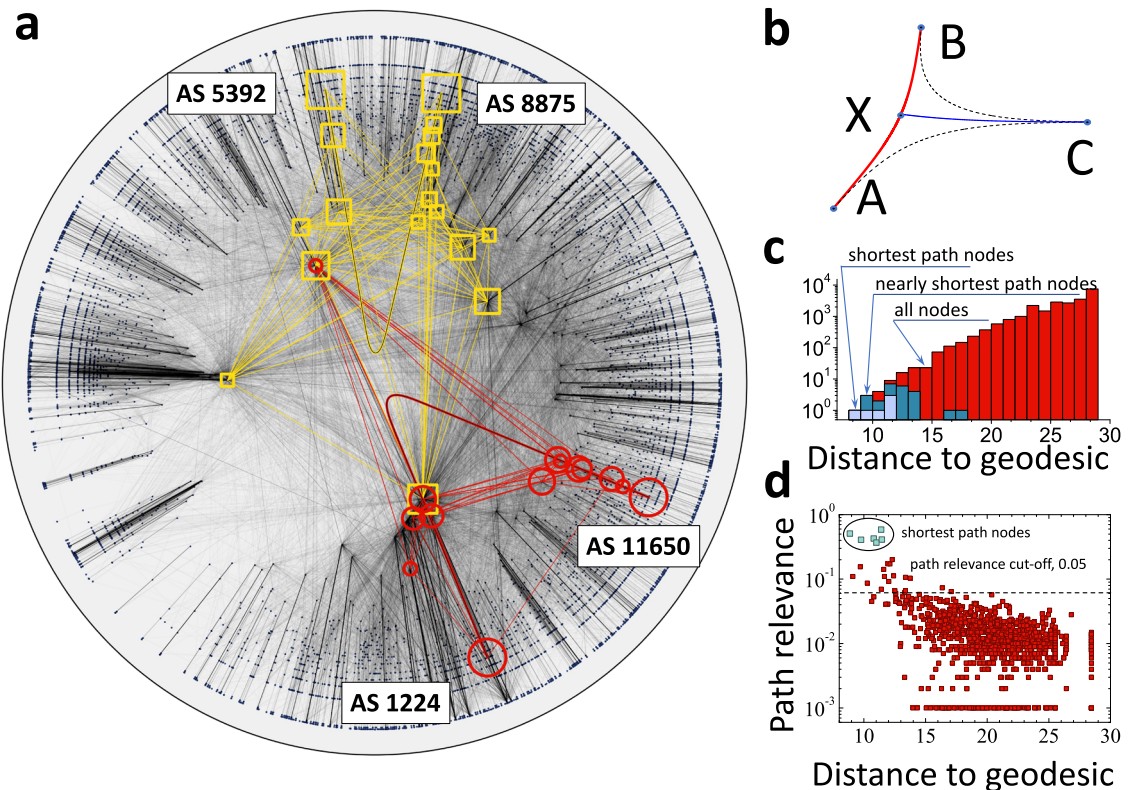

**Fig. 1 | Latent geometry uncovers shortest and nearly shortest paths in the Internet at the Autonomous System level. a** Hyperbolic map of the Internet at the Autonomous System (AS) level. See Section SV for data collection and hyperbolic mapping details. The latent space is the 2-dimensional hyperbolic disk and each point corresponds to an Autonomous System. Yellow squares and red circles highlight ASes corresponding to communication paths between AS5392-AS8875 and AS1224-AS11650 pairs. Shown are the nodes with path relevance exceeding 0.1, see Methods. Sizes of squares and circles are larger for nodes with higher path relevance. **b** Schematic distance from point $C$ to geodesic $\gamma(A, B)$ drawn between points $A$ and $B$. **c** The distribution of distances to the $\gamma(AS5392, AS8875)$ geodesic from (light blue) shortest path nodes, (dark blue) nodes with path relevance larger than 0.05, and (red) all Internet nodes. **d** Path relevance as a function of distance to the $\gamma(AS5392, AS8875)$ geodesic for all ASes.

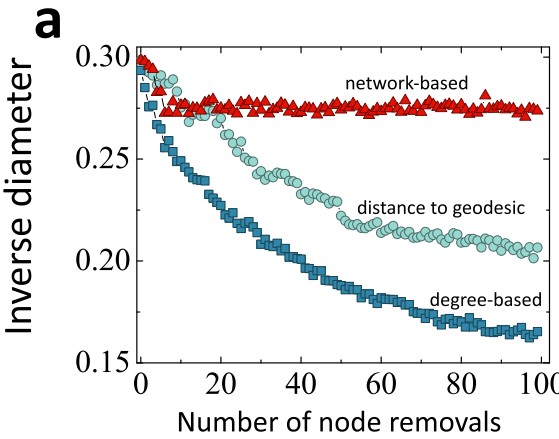
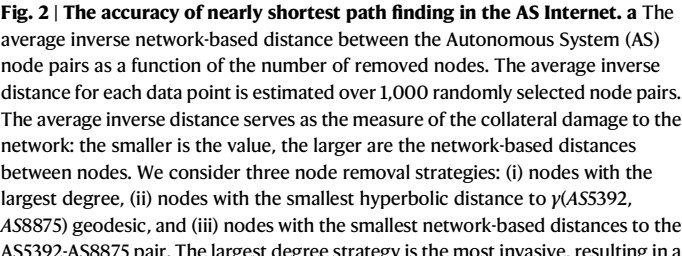
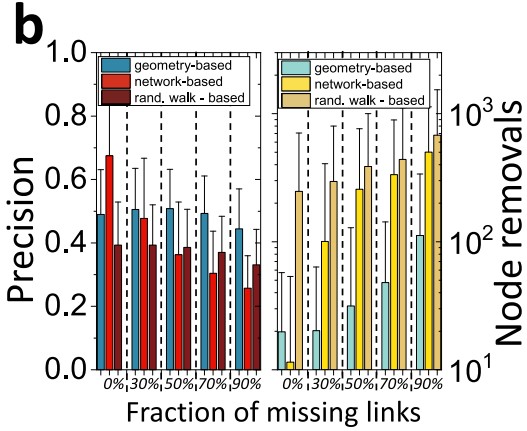

**Fig. 2 | The accuracy of nearly shortest path finding in the AS Internet. a** The average inverse network-based distance between the Autonomous System (AS) node pairs as a function of the number of removed nodes. The average inverse distance for each data point is estimated over 1,000 randomly selected node pairs. The average inverse distance serves as the measure of the collateral damage to the network: the smaller is the value, the larger are the network-based distances between nodes. We consider three node removal strategies: (i) nodes with the largest degree, (ii) nodes with the smallest hyperbolic distance to $\gamma(AS5392,$ $AS8875)$ geodesic, and (iii) nodes with the smallest network-based distances to the AS5392-AS8875 pair. The largest degree strategy is the most invasive, resulting in a

nearly two-fold decrease of the average inverse network-based distance upon removing 100 largest degree nodes. This result is consistent with earlier fundings on attack tolerance of complex networks[40]. **b** The accuracy of distance to geodesic in finding nearly shortest path nodes in the incomplete AS Internet network. The distance to geodesic strategy is juxtaposed against the network-based strategy and random walk-based strategies, see Methods. We evaluate the average precision scores and the average number of node removals needed to disrupt node pairs of interest. Each data bar is the average over 1000 randomly selected node pairs, and error bars display one standard deviation. Note that the relative performance of the distance to geodesic increases as the fraction of missing links increases.

We calculated distances from all network nodes to the hyperbolic geodesic connecting the AS5392-AS8875 node pair, finding that all 6 shortest path nodes are among the 12 closest to the geodesic nodes, see Fig. 1**c**. Although not part of the original shortest path, the remaining 6 closest to the geodesic nodes may form alternative shortest paths if the Internet topology is perturbed. To verify this claim, we calculated for each node its path relevance, which we defined as the probability to be on the shortest path in case network links are removed uniformly at random with probability $q = 0.5$. We found, see Fig. 1**d**, that node path relevance is anti-correlated with the distance to geodesic, and the closest to the geodesic nodes are characterized by the largest path relevance values, confirming our claim. The anti-correlation between path relevance and the distance to geodesic also indicates that distance to geodesic is capable of identifying not only the shortest path nodes but also nodes that may belong to a shortest path if the network topology is perturbed.

Before further explaining our results, we need to discuss some technical challenges and provide some rigorous definitions. First, many real networks are characterized by the small-world property: network-based distances between the nodes scale logarithmically[9] or even sub-logarithmically[10] as a function of network size $N$. Similarly, the sets of the shortest path nodes are extremely small compared to the network size, making the shortest path classification problem extremely unbalanced[11]. Second, from the perspective of dynamic processes, such as communication or viral spreading, the propagation along nearly shortest paths is not much worse than the propagation along the shortest paths. These two observations motivated us to consider nearly shortest path nodes along with the shortest path nodes. To do so, we combined both entities under the umbrella of the path relevance metric, Fig. 1**d**, and defined nearly shortest path nodes as nodes with path relevance values exceeding 0.05. Nearly shortest path nodes address both challenges. Indeed, they contain not only the original shortest paths but also nodes on slightly less optimal paths. Also, the sets of the nearly shortest path nodes are larger than those of the original shortest paths, reducing the classification imbalance.

To quantify the accuracy of the identification of nearly shortest path nodes, we use two metrics. The first metric is the statistical precision score, defined as the fraction of true nearly shortest path nodes.

The second metric is the number of node removals necessary to disrupt all paths between the pair of nodes of interest, Section SVIII. While indirect, the second metric provides an insight into how effective a path identification method of interest is in finding possible path deviations.

We compare the accuracy of the distance to geodesic metric $d(C, \gamma(A, B))$ to its network-based and random walk-based counterparts. In analogy to distance to geodesic, we define network-based metric $d_{nb}(C|A, B) = \ell_{A,C} + \ell_{C,B}$, which is the sum of the shortest path distances from node $C$ to path endpoints $A$ and $B$. In addition, we consider a random walk hit frequency $d_{rw}(C|A, B)$ and the average commute time metric $d_{comm}(C|A, B)$, see Methods and Section SVIII.

Since shortest paths are known to traverse the most connected nodes in networks[12], it is tempting to use node degree as an alternative path relevance metric. The node degree is, however, of limited practical use since it is usually hard, if not impossible, to control large-degree nodes. The most connected Internet ASes are the largest Internet Service or Content Providers that are rarely affected by adverse events[13]. Proteins with large numbers of interaction partners are hard to manipulate due to the increased possibility of side-effects[14]. Large degree nodes are usually shared by many communication paths, as quantified by the betweenness centrality[15], and manipulating them will affect not only the path of interest but also all other paths in the network. As seen in Fig. 2**a**, removing nodes with the largest degree values affects the lengths of all paths, decreasing the average inverse network diameter.

**Finding nearly shortest paths in incomplete networks**

To quantify the alignment of nearly shortest paths, we conducted a series of path-finding experiments on the AS Internet, the PGP web of trust, and the similarity-based human PPI network with varying fractions of missing links. The similarity-based PPI network is the derivative of the traditional PPI network. In our construction, two proteins are linked if they have a statistically significant number of common interaction protein partners, see Section SVI and Supplementary Data 2. Since two proteins with common interaction partners can be interpreted as similar, we refer to the resulting network as the similarity-based PPI network. Nodes in the PGP network are certificates

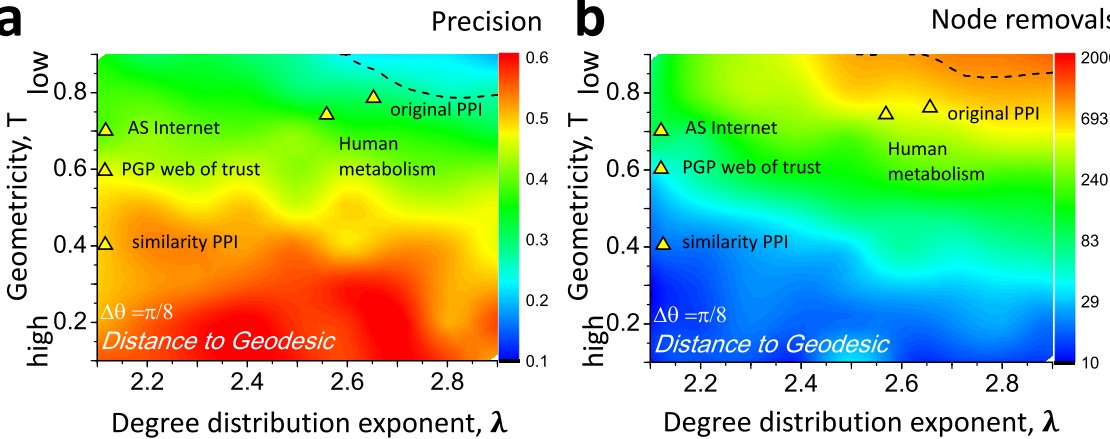

**Fig. 3 | The accuracy of nearly shortest path finding in synthetic networks incomplete synthetic networks.** Synthetic networks are constructed as Random Hyperbolic Graphs (RHGs) of $N = 5,000$ nodes, average degree $\langle k \rangle = 10$, and variable degree distribution, $\lambda \in (2, 3)$, and geometricity $T \in (0, 1)$ parameters. After the construction of each network, we remove each of its links with probability $p = 0.5$. The heatmaps consist of $9 \times 9 = 81$ points, each point corresponding to the average of 100 randomly selected node pairs separated by the $\Delta\theta = \frac{\pi}{8}$ angle in $\mathbb{H}^2$. For technical details and other angles, see Section SVIII. Panel **a** displays precision scores, while panel **b** displays the number of node removals needed to disconnect all paths between the node pair of interest. Note that the path-finding accuracy is nearly independent of degree distribution. The highest path-finding accuracy is achieved at temperature $T = 0$, when the geometricity of RHGs is the strongest and decreases as $T$ increases. Marked on panels **a**, **b** are inferred parameters of real networks. These networks are the Internet at the autonomous system (AS) level ($\lambda = 2.1$, $T = 0.7$) similarity protein-protein interaction (PPI) network ($\lambda = 2.1$, $T = 0.4$), and the pretty-good-privacy (PGP) web of trust PGP web of trust ($\lambda = 2.1$, $T = 0.6$) that are studied in this work, the network of human metabolic interactions (Human metabolism, $\lambda = 2.55$, $T = 0.65$), Ref. [41], and the network of protein-protein interactions (original PPI, $\lambda = 2.65$, $T = 0.7$), Ref. [4].

of public PGP keys and links are trust relationships between them, see Section SVII. In all experiments, we first computed the true nearly shortest path nodes using the original network. We then removed randomly selected links and tried to identify nearly shortest path nodes on incomplete networks. In the case of the distance to geodesic, we obtained hyperbolic maps of the incomplete networks. For every node pair $A$-$B$ of interest, we then determined corresponding hyperbolic geodesic $\gamma(A, B)$ and then used distance to geodesic $d(C, \gamma(A, B))$, Eq. (1), to quantify the proximity of other network nodes to $\gamma(A, B)$. For comparison, we computed the alternative network-based and random walk based scores $d_{\text{nb}}(C|A, B)$, $d_{\text{rw}}(C|A, B)$, and $d_{\text{comm}}(C|A, B)$ directly on incomplete networks.

We observe that the accuracy of the distance to geodesic only decreases mildly as the fraction of missing links increases, Fig. 2**b**, Fig. S12, and Fig. S13. This result is in sharp contrast with the accuracy of the network-based method, which decreases fast as the rate of the missing links increases. While both the network-based method and the distance to geodesic yield comparable results on complete networks, in cases of substantially incomplete networks, 70% and 90% of the missing links, the precision of the distance to geodesic is approximately twice than that of the network-based method, Fig. 2**b**. Similarly, in the case of the 70% missing link rate, it takes six times more node removal steps, on average, to disrupt all paths with the network-based ranking, compared to that of the distance to geodesic ranking, Fig. 2**b**. This is the case since network-based path finding methods are extremely sensitive to missing network data, Section SIV. Traditional shortest path finding methods, like the classical Dijkstra algorithm[16], iteratively explore network-based neighborhoods of the path end-nodes. As a result, network-based methods rely on the accuracy of the network data and are doomed to fail if some of the network data is missing. Latent-geometric maps of networks, on the other hand, provide an effective mean-field image of the network, which is not very sensitive to uniformly missing network data[17], Section SIV. As a result, the distance to geodesic is reliable even in the case of substantially incomplete networks, where the number of missing links exceeds that of known links. While the random-walk-based metrics $d_{\text{rw}}(C|A, B)$, and $d_{\text{comm}}(C|A, B)$ are less sensitive to missing data compared to

network-based metrics, their overall accuracies appear to be subpar than that of the distance to geodesic, Figs. 2, 3, S12-S15.

In practice, incomplete networks contain not only missing but also spurious links. Therefore, it is logical to ask if the latent geometric localizations of nearly shortest paths are strong enough when spurious links are present. To answer this question we repeated our path-finding experiments for various rates of missing links $q \in [0, 0.9]$ when a constant fraction of 10% of spurious links were added uniformly at random, arriving at the same conclusions, Fig. S14. We also found that distance to geodesic is accurate in finding nearly shortest paths when only spurious links are present, Fig. S15. When only spurious links are present, however, distance to geodesic offers accuracy inferior to that of the network-based methods. This is the case since randomly added spurious links preferentially connect small degree nodes in scale-free networks and, as a result, rarely affect original nearly shortest paths that tend to pass mid and high-degree nodes, see Section SVIII.

## Identifiability limits for nearly shortest paths

To explore the limits of the distance to geodesic metric in the identification of shortest path nodes, we conducted experiments on incomplete random hyperbolic graph (RHG) models[18,19]. RHG models are obtained by sprinkling network nodes into a 2-dimensional hyperbolic disk $\mathbb{H}^2$ and connecting node pairs with distance-dependent probabilities, Section SIII. RHGs are used as null models in hyperbolic network embedding methods and also allow the generation of synthetic networks with scale-free degree distribution $P(k) \sim k^{-\lambda}$ with variable exponent $\lambda \in (2, 3)$ and variable degree of geometricity that is controlled by the temperature parameter $T \in [0, 1]$. In the limiting case of $T = 0$, network links are only allowed between node pairs at small latent distances in $\mathbb{H}^2$. In this case, network geometricity is the strongest since all network links are short-range. As $T$ increases, connections at larger distances are allowed with increasing probabilities, leading to weaker network geometricity, Section SVIII.

To identify paths in RHGs using distance to geodesic, we first erased their original node coordinates in $\mathbb{H}^2$ and then re-learned them using the HL embedder[17]. Our path finding experiments for incomplete RHGs suggest that the accuracy of the distance to geodesic is nearly independent of the degree distribution exponent $\lambda$ and strongly

depends on network geometricity, as quantified by temperature $T$, Fig. 3**a**, **b**. As $T$ increases, the latent-geometric path finding accuracy decreases and becomes comparable to that of the network-based method, Fig. 3a, **b** and Figs. S8**a**, S9**a**, S10**a**, S11**a**. For comparison, we obtained similar heatmaps for the network-based distance, Figs. S8**b**, S9**b**, S10**b**, S11**b**, observing the distance to geodesic is superior to the network-based distance nearly in the entire range of $\lambda$-$T$ parameters, except for the largest $T$ values, as shown by the dashed lines in Fig. 3**a**, **b**.

Distance to geodesic can be invaluable in the analysis of paths in incomplete networks. One family of applications, to this end, concerns the validation of existing communication paths. The validation of communication paths is much needed in distributed communication networks, such as the Internet. Another family of applications stems from the ability of distance to geodesic to find alternative nearly shortest paths. The latter task of finding possible path deviations is relevant not only in communication networks but also in cellular pathways, as we discuss below.

### Assessing the integrity of routing paths with distance to geodesic

Autonomous Systems (ASes) comprising the Internet are independent organizations. Hence, the information on how to reach devices in another AS is not readily available to them. This reachability information is disseminated by the Border Gateway Protocol (BGP)[20]. BGP belongs to the family of path-vector routing protocols: ASes share with their neighbor ASes paths to various destinations known to them. BGP routing is based on trust: ASes accept routing paths advertised by their neighbors without strict integrity tests, Fig. 4**a**. With the increasing number of recent cyberattacks and routing instabilities, path integrity checks are becoming much desired[21]. One class of cyberattacks is the BGP prefix hijacking. During a BGP prefix hijack, an AS either claims ownership of IP address prefixes that are owned by other ASes or falsely announces that it can provide transit to a prefix or a set of prefixes. This attack can compromise the affected data flows by either exposing them to malicious actors or simply misrouting them[21], Fig. 4**a**.

While interdomain Internet routing paths are known to be longer than shortest paths due to many factors, including AS business relationships[22], the extent of their inflation, as found in Ref. [23], places them into the category of nearly shortest paths. Since the latter tend to align along geodesics in the Internet hyperbolic representation, we propose that the integrity of Internet routing paths can be assessed by charting them in the latent hyperbolic space. We expect genuine routing paths to localize in the vicinity of the hyperbolic geodesic connecting path endpoints, Fig. 4**b**. Fake routing paths, on the other hand, are expected to significantly deviate from corresponding geodesics, allowing for their detection, Fig. 4**c**.

To quantify the alignment of routing paths, we propose the hyperbolic stretch $D_{\Omega(A, B)}$, which we define as the maximum normalized distance from path $\Omega(A, B)$ to the geodesic curve $\gamma(A, B)$ connecting path endpoints, $D_{\Omega(A,B)} = \max_{C \in \Omega(A,B)} \frac{d(C, \gamma(A,B))}{d(A,B)}$, Section SV. We studied three recent BGP instability events involving MainOne Telecom in Nigeria, Malaysia Telecom (MT), and Rostelecom (PJSC), respectively, Section SV. We analyzed routing paths involving the affected ASes that were advertised both before and during each instability event. As seen from Fig. 4**d** and Fig. S4, stretches of routing paths advertised during the hijack events are significantly larger than those before the event, suggesting that ASes may use the hyperbolic stretch to reject questionable routing paths, Section SV. In contrast to network-based methods, ASes do not need to know the exact AS-level network to assess routing path integrity. To compute the hyperbolic stretch, one only needs to know the hyperbolic coordinates of ASes constituting the routing path of interest. Our results indicate that AS coordinates can be computed with sufficient accuracy even when a large fraction of network links is unknown.

### Exploring the neighborhoods of cellular pathways in human PPI network

Complex cellular processes and many diseases involve multitudes of genes and their products that are organized into molecular pathways. It is also not uncommon for the same pathway to be linked to several diseases or be intrinsically involved in more than one molecular process. One of the key questions arising from the point of view of network biology is the functional relationship between the pathway genes and the genes located in their proximity.

Are cellular pathways akin to communication paths? There is no clear-cut answer to this question: different from the communication paths, cellular pathways often have no single origin and destination. Neither do cellular pathways conduct traffic. While the communication paths have a clear objective to be optimal, there is no such a requirement for cellular pathways, although there is an expectation that the cellular pathways evolved to become optimal[24]. In the light of these differences, an intriguing finding of our work is that some cellular pathways align along the geodesic curves when drawn on the hyperbolic representations of PPI networks, see Fig. 5**a** and Fig. S6.

We studied the ubiquitin-proteasome (UPP), the transforming growth factor beta (TGFb), and the cell cycle pathways. For each pathway, we identified its localizing hyperbolic geodesic curve using the least squares fitting for pathway proteins, Section SVI. We have observed that the two halves of the geodesic curves in these pathways often naturally split the proteins associated with the pathways to the functionally related subsets of proteins. For the UPP pathway, Fig. 5**a**, we found that proteins in the geometric vicinity of the fitted geodesic are associated with E2 and E3 enzyme classes. We found that the larger geodesic branch at 10 o'clock is associated with E2 and E3 classes but not with E1. The 12 o'clock branch, on the other hand, is exclusively associated with the E3 class, Fig. 5**a**.

To further explore the analogy between cellular pathways and communication paths, we asked if other genes that are functionally similar to the ones in the geodesic but lie outside of it can be found using their proximity to the geodesic. To answer this question, we considered 100 genes in the latent-geometric proximity to the ubiquitin-proteasome pathway (UPP), as quantified by the distance to the fitted geodesic, Eq. (1). We used DAVID tools[25], a bioinformatics framework specifically designed to provide systematic functional analysis for a large group of genes, to functionally cluster groups of the proximal genes independently from the genes in the geodesic cluster, finding 6 major clusters, Section SVI. As expected, the most highly ranked cluster contained the terms related to ubiquitination, cluster 1, Fig. S7**a**, and Supplementary Data 3. Interestingly, the other clusters include genes associated with the immune response signaling pathway, cluster 2, Fig. S7**b**, viral infections, clusters 3, Fig. S7**c** and 4, Fig. S7**d**, pathways associated with several types of cancer, clusters 3 and 4, as well as zinc fingers, cluster 5, Fig. S7**e**, and DNA repair, cluster 6, Fig. S7**f** and Supplementary Data 3. Each of the functional groups is naturally associated with ubiquitination[26–30]. Indeed, ubiquitination is a key mechanism regulating signal transduction and mediating both innate and adaptive immune responses[26]. On the other hand, the principal role of protein containing zinc finger domains in ubiquitination has emerged only recently[31–34].

Our observations indicate that the UPP, TGFb, and cell cycle pathways are organized similarly to communication paths. Not only are these pathways aligned along geodesic curves, but other genes in the latent geometric vicinity to them appear to be functionally related. To complete the analogy, it is tempting to ask whether perturbations in protein interaction networks result in perturbed pathways, which follow the same geodesic. While we do not have the answer to this question yet, we hypothesize that two likely scenarios can happen when replacing a malfunctioning gene, e.g., due to a deleterious mutation. The first scenario is when the new gene is a parlor of the replaced gene, and the reason that it was not in the original pathway is

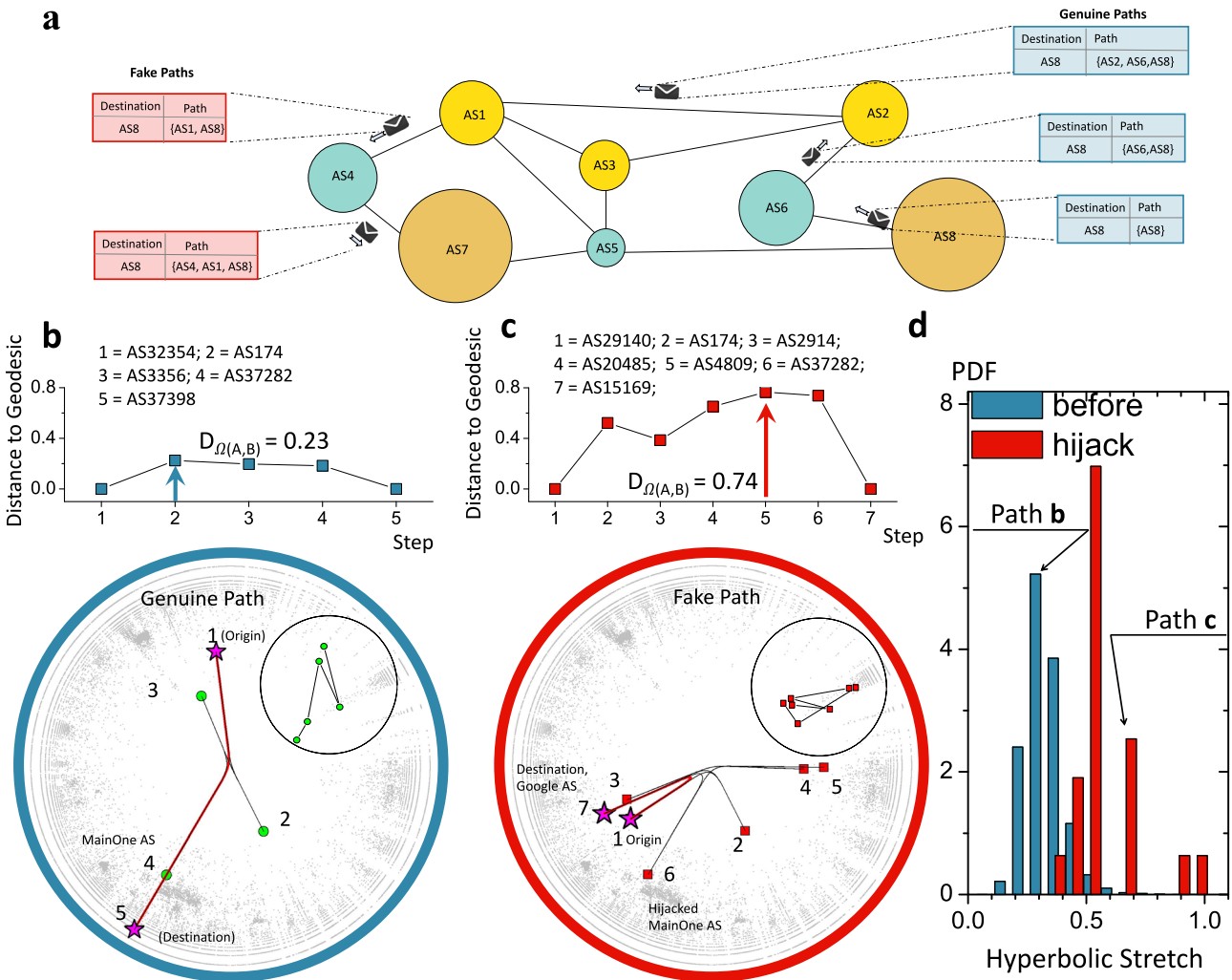

**Fig. 4 | Anomaly detection in interdomain Internet routing. a** Interdomain routing at a glance. Shown is toy network of the Internet at the Autonomous System (AS) level, where nodes are autonomous systems and links are data transfer agreements. Shown on the right-hand side is the distributed calculations of paths to AS8. The generation of paths on the left-hand side is an example of routing instability. AS1 falsely claims the direct connection to AS8. This information is propagated by the border-gateway-protocol (BGP) first to AS4 and then to AS7. As a result, AS4 and AS7 have fake path information to AS8. Routing paths following the hijack of the Google AS prefixes by the MainOne AS, see Section SV. **b** An example of a routing path traversing Nigerian Internet Service Provider MainOne (AS37282) announced before the prefix hijack. Green circles are ASes constituting the path; the solid red line is the hyperbolic geodesic connecting the origin-destination pair. Distance to geodesic for every node constituting the routing path is shown above the hyperbolic map. The hyperbolic stretch of the BGP path, Eq. (1), is the largest

normalized distance to geodesic from routing path nodes, $D_{\Omega(A, B)} = 0.23$. **c** An example of a routing path announced during the prefix hijack of MainOne AS. MainOne announced a direct connection to Google AS (AS15169), leading to a large-scale cascade of false BGP path announcements. Shown is one of these false BGP paths originating at AS29140, traversing the MainOne AS (AS37282) and ending at the Google AS (AS15169). Distance to geodesic for every node constituting the routing path is shown above the hyperbolic map. The hyperbolic stretch of the routing path is $D_{\Omega(A, B) = 0.74}$, indicating that the BGP path is less conformal to the latent-geometric geodesic, compared to the one in panel **b**. Red squares are ASes constituting the path; the solid red line is the hyperbolic geodesic connecting the origin-destination pair. **d** PDFs of hyperbolic stretches for BGP paths announced (blue) before and (red) during the MainOne AS prefix hijack. BGP paths announced before the hijack are characterized by smaller hyperbolic stretch values than those announced during the hijack.

that it performs the function less efficiently than the original gene. In this case, the new pathway will be longer compared to the original one. The second scenario is when the new gene corresponds to an alternatively spliced variant that could perform the function in the same manner or even more efficiently but is under-expressed compared to the original gene that is the primary spliced variant. In this case, if the original gene is removed, the new node will become the new primary splice variant, with a much higher relative expression due to the absence of the old primary splicing variant. As a result, we expect the new pathways to be of the same length or even shorter.

## Discussion

There is no one-size-fits-all solution to the shortest path problem. In order to identify shortest path nodes in a partially known network, one

needs to know both the mechanisms of network formation and the character of missing data. Distance to geodesic, in this respect, assumes that link formation in the network is captured by its latent geometry, and unknown links are missing uniformly at random.

The first condition is a must: one cannot expect to identify shortest paths in non-geometric networks using geometric methods. More than that, the embedding space must agree with network topology, and learned embeddings must be of sufficiently high accuracy. How do we know if the network of interest has effective hyperbolic geometry? It is well known that spatial network models built in hyperbolic spaces are sparse, self-similar, have strong clustering coefficients, and are characterized by scale-free degree distributions[3]. These topological properties can be regarded as necessary conditions of network hyperbolic geometricity. Many real networks, including the ones in this study, have

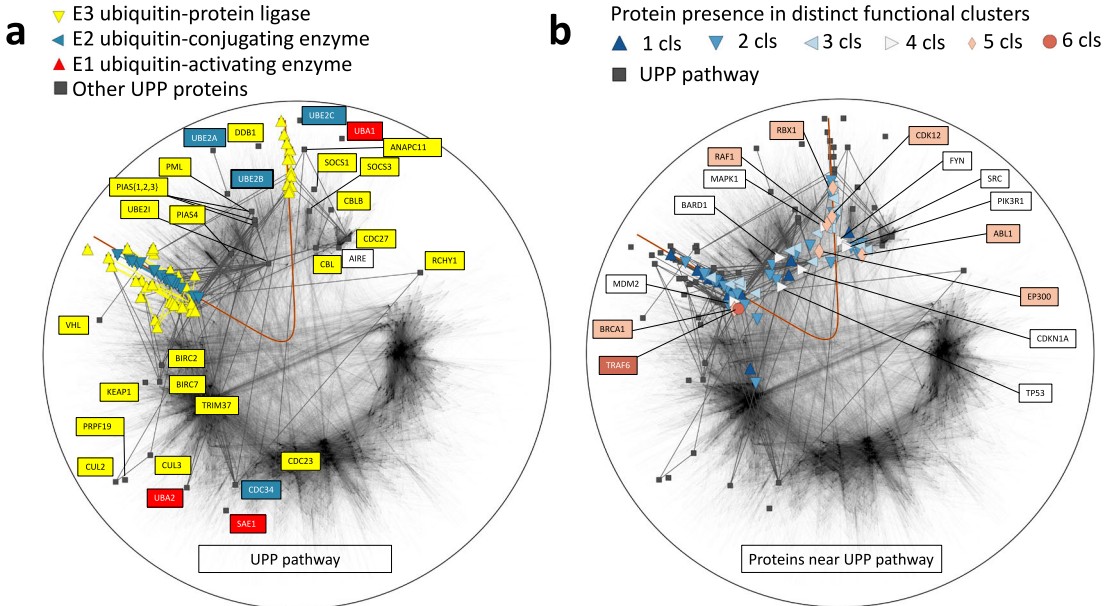

**Fig. 5 | Latent-geometric localization of the ubiquitin-proteasome pathway (UPP) pathway. a** the hyperbolic map of the similarity-based human protein-protein interaction (PPI) network and the UPP pathway. Pathway proteins are colored according to their functional groups, background nodes are non-pathway proteins comprising the network. E1, E2, and E3 are the three classes of enzymes associated with ubiquitination: E1 and E2 correspond to ubiquitin-activating and ubiquitin-conjugating enzymes, respectively, while E3 correspond to ubiquitin-protein ligases. The solid line displays the hyperbolic geodesic fitting the pathway proteins. Panel **b** depicts 6 clusters of proteins in the latent-geometric vicinity of the UPP pathway geodesic. Proteins are colored based on the number of clusters they belong to. Black squares depict UPP pathway proteins. See also Fig. S7, which depicts each of the 6 clusters separately.

been shown to meet these necessary conditions. Whether the same topological properties are also sufficient conditions of network hyperbolic geometricity is an open research problem. The first result in this direction is the equivalence between network ensembles with fixed clustering and expected degree and random geometric graphs on a straight line[35]. The extension of this equivalence result to random geometric graphs in hyperbolic spaces is not easy[3].

In this work, we demonstrated that hyperbolic maps of incomplete networks can be used to find shortest and nearly shortest path nodes. Can *Euclidean* spaces also be used in path-finding tasks? To answer this question, we employed two classical *Euclidean* embedding methods, Node2Vec[36] and DeepWalk[37]. Our experiments on the incomplete AS Internet network suggest that these embedding methods, in their original formulation, are not capable of finding nearly shortest paths with sufficient accuracy, see Fig. 6. We do observe, however, that higher dimensionality $D$ in *Euclidean* embedders improves the path-finding accuracy, suggesting that *Euclidean* spaces of sufficiently high dimensionality combined with properly tuned embedding parameters may result in embeddings suitable for path-finding tasks. The observed effect of higher *Euclidean* dimensionality improving path-finding accuracy is hardly surprising: higher dimensionality makes *Euclidean* spaces conceptually closer to their hyperbolic counterparts, which grow exponentially in any dimension. Returning back to hyperbolic embeddings, we asked how accurately does one need to learn node coordinates to be able to identify nearly shortest paths? To answer this question we added synthetic noise of variable magnitude to learned hyperbolic coordinates of the incomplete AS Internet network. As expected, the path-finding accuracy does decrease as the noise magnitude increases. Yet, the observed moderate decrease rate suggests that the latent-geometric path-finding approach can tolerate small inaccuracies in learned coordinates, Fig. 6. Somewhat more surprising, we observed a sharp drop in path-finding accuracy as synthetic noise magnitude exceeds a certain threshold, suggesting a possible phase transition in latent-geometric path predictability.

The second assumption – of uniformly missing links–could probably be relaxed in future research. A relatively straightforward step in this direction is to assume that the probabilities of missing and spurious links are also certain functions of latent distances between the nodes. A more general framework for arbitrary missing and spurious links, however, may prove substantially more challenging.

In summary, we established that latent-geometric geodesics serve as fairways for shortest and nearly shortest path nodes in geometric networks. Nodes in the vicinity of these geodesics are likely to lie on shortest paths or may become shortest path nodes if the network topology is perturbed. We found that distance to geodesic offers a reliable way to find nearly shortest path nodes even in substantially incomplete networks. Our finding can be either a curse or a blessing, depending on the circumstances. One could exploit the geometric localization of shortest paths to disrupt or eavesdrop on communication paths of interest. On the other hand, the knowledge of geodesic fairways may help identify alternative optimal paths and rule out inefficient or fraudulent paths in communication networks.

## Methods

### Path relevance and nearly shortest path nodes

We define the relevance of node $C$ to paths connecting nodes $A$ and $B$ as the probability that $C$ belongs to the shortest path connecting $A$ and $B$ if network links are removed uniformly at random with probability $q$. The path relevance metric allows to identify not only the original shortest path nodes but also nodes that may become shortest either if the network topology is perturbed or if the original shortest path becomes unavailable. In all experiments, we set the missing link probability to $q = 0.5$. We define nearly shortest path nodes connecting nodes $A$ and $B$ as the nodes with the $A$-$B$ path relevance exceeding 0.05.

### Distance to geodesic

Distance between two points $\{r_i, \theta_i\}$ and $\{r_j, \theta_j\}$ in the 2-dimensional hyperbolic disk $\mathbb{H}^2$ is given by the hyperbolic law of cosines

$$\cosh \zeta d_{ij} = \cosh \zeta r_i \cosh \zeta r_j - \sinh \zeta r_i \sinh \zeta r_j \cos \Delta \theta_{ij}, \quad (2)$$

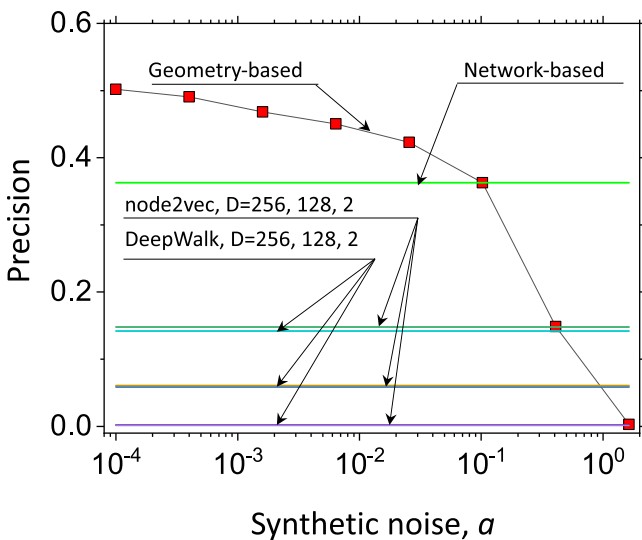

**Fig. 6 | Noise tolerance and comparison to other embedding methods.** We analyze the tolerance of distance to geodesic to node coordinate uncertainties. We consider the hyperbolic map of the incomplete network of the Internet at the Austonomous System (AS) level at the $q = 0.5$ missing link rate. We plot the precision of distance to geodesic in finding nearly shortest path nodes as a function of synthetic noise magnitude $a$, which we add to learned node coordinates: $\hat{\theta}_i = \theta_i + a2\pi X_i$, $\hat{r}_i = r_i + aRY_i$, $\{X_i, Y_i\} \leftarrow U(0, 1)$. We compare the precision of distance to geodesic to the network-based method and two popular *Euclidean* embedding algorithms node2vec[36] and DeepWalk[37] of different dimensionality $D$. Each simulation is the average over 1000 randomly chosen node pairs. In cases of node2vec and DeepWalk, we rank nodes based on their distance to *Euclidean* geodesic drawn between the path endpoints. For each *Euclidean* embedding algorithm, the precision increases as a dimensionality $D$ increases. Note that both node2vec and DeepWalk are less accurate than the network-based metric.

and for sufficiently large $r_i$ and $r_j$ values is closely approximated by $\mathbf{x}_{ij} = r_i + r_j + \frac{2}{\zeta}\ln\left(\sin(\Delta\theta_{ij}/2)\right)$, see Section SII.

Distance from point $C$ to geodesic $\gamma(A, B)$ as the shortest distance from $C$ to any point on $X \in \gamma(A, B)$:

$$d(C, \gamma(A, B)) = \min d(C, X), \qquad (3)$$

$$\text{s.t. } X \in \gamma(A, B) \qquad (4)$$

The distance to the hyperbolic geodesic $d(C, \gamma(A, B))$ is closely approximated by Eq. (1), see Section SII.

### Alternative path-finding metrics

In our work, we compare the accuracy of distance to geodesic to several alternative metrics.

Network-based metric: $d_{nb}(C|A, B) = \ell_{A,C} + \ell_{C,B}$, where $\ell_{X,Y}$ is the shortest path distance between nodes $X$ and $Y$. $d_{nb}(C|A, B)$ is minimized when $C$ lies on a shortest path between $A$ and $B$. The larger $d_{nb}(C|A, B)$ the less the relevance of $C$ to paths between $A$ and $B$.

Random walk-based metric 1: $d_{comm}(C|A, B) = n(A, C) + n(C, B)$, where $n(X, Y)$ is the average commute time between $X$ and $Y$. $n(X, Y) \equiv m(X|Y) + m(Y|X)$, where $m(X|Y)$ is the mean first passage time from $Y$ to $X$. $n(A, C)$ can be computed efficiently with the pseudoinverse of the network's Laplacian matrix, Ref. 38, see Section SVIII. We were able to commute $d_{comm}(C|A, B)$ for the similarity-based PPI and the PGP web of trust networks but not for the Internet due to its large size.

Random walk-based metric 2: $d_{rw}(C|A, B)$ is the random walk hit frequency. To compute this metric, we initiate $M$ independent random walks from nodes $A$ and $B$ of fixed length $D$, counting the number of

times random walks visit $C$ from $A$ and $B$. In our simulations we use $M = 1000$ and $D = 20$.

### Network mapping

Network mapping or embedding into a latent space $\mathcal{M}$ is a procedure of determining the coordinates of nodes constituting the network in this space. In this work we map AS Internet and the similarity-based protein interaction network to the 2-dimensional hyperbolic disk using the HL Embedder algorithm[17]. Similar to other hyperbolic embedders, HL embedder maps network nodes to points $\{r_i, \theta_i\}$, $i = 1,...,N$, in a hyperbolic disk $\mathbb{H}^2$ by maximizing the posterior probability $\mathcal{L}(\{r_i, \theta_i\}|a_{ij})$ that the network with the adjacency matrix $a_{ij}$ has given node coordinates and is generated as the RHG model, Section SIII. By the Bayes' rule, $\mathcal{L}(\{r_i, \theta_i\}|a_{ij}) \propto \frac{\mathcal{L}(a_{ij}|\{r_i, \theta_i\})}{\text{Prob}(\{r_i, \theta_i\})}$, where $\mathcal{L}(a_{ij}|\{r_i, \theta_i\})$ is the likelihood that the network $a_{ij}$ is generated as the RHG, given node coordinates $\{r_i, \theta_i\}$, and the $\text{Prob}(\{r_i, \theta_i\})$ is the prior probability of node coordinates generated by the RHG. Since links $\{ij\}$ in the RHG are established independently with probabilities depending on hyperbolic distances $\{\mathbf{x}_{ij}\}$ between the nodes, $\mathcal{L}(a_{ij}|\{\mathbf{x}_i\}) = \prod_{i<j}[p(x_{ij})]^{a_{ij}}[1 - p(x_{ij})]^{1-a_{ij}}$. The HL embedder is freely available at the github repository[39].

### Reporting summary

Further information on research design is available in the Nature Portfolio Reporting Summary linked to this article.

## Data availability

Data pertaining to the analysis of the AS Internet and the PPI network are included as Supplementary Data 1 and Supplementary Data 2, respectively.

## Code availability

The hyperbolic embeddings are performed using the Hyperbolic Embedder, which is hosted by the Bitbucket repository[39].

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

## Acknowledgements

The authors thank to I. Voitalov, H. Hartle, W.L.F. van der Hoorn, P.F.A. Van Mieghem, and D. Krioukov for stimulating discussions. This work was supported by the U.S. Defense Threat Reduction Agency. Additional support was provided by the U.S. Army Corps of Engineers under FLEX project on Compounding Threats. MK was additionally supported by NSF grant IIS-1741355, ARO grants W911NF-16-1-0391 and W911NF-17-1-0491, and the Dutch Research Council (NWO) grant OCENW.M20.244. DK is supported by NIH grant R01LM014017. The views and opinions expressed in this article are those of the individual authors and not those of the U.S. Army, or other sponsor organizations.

## Author contributions

M.K., A.E., D.L.A., D.K., and I.L. designed the research. M.K., A.G., A.E., H.C., and D.K. performed the research, M.K., A.G., A.E., H.C., D.A.E., D.L.A, D.K., and I.L. discussed research results and wrote the manuscript, M.K. was the lead writer of the manuscript.

## Competing interests

The authors declare no competing interests.
