## [Peer Review File · Nature Communications]

REVIEWER COMMENTS

Reviewer #1 (Remarks to the Author):

The main point of this work is that if we map the nodes of a network into a latent geometric (hyperbolic to be precise) space, then the shortest paths in that space are often reasonably close (but not exactly -- and sometimes with substantial errors) to the shortest paths in the "native" network. Additionally, the paper makes the case that if we do not know the network accurately, in the sense that many real links may be missing from our inferred network, we can still estimate "almost shortest paths" reasonably well. The paper presents some results and applications of this approach on an Internet AS-level topology and on a similarity-based protein-protein interaction (PPI) network for Homo Sapiens.

The paper is well written, clear in its goal and methodology, and quite thorough and methodic in its scientific approach.

On the negative side, I do not think it represents a significant contribution -- something that could have either practical impact or that it adds important new knowledge in the study of complex networks. I will explain the reasons for this negative assessment next:

1. As the paper briefly admits in the Discussion, this entire approach is based on the assumption that the network has been formed in a way that is consistent with a well-characterized geometric embedding. The authors correctly write: "one cannot expect to identify shortest paths in non-geometric networks using geometric methods." I would argue that it is actually more challenging than that: we need to know well the underlying geometry of the network.

Now the real issue here is that if the network is inaccurately known, which is the main "application focus" of this work, we may not be able to characterize that underlying geometry -- simply because we do not know about many links. If the network is inaccurately inferred, then even if we think that it is consistent, say, with hyperbolic embeddings, in reality it may not do so.

So how can we be sure that the paths we compute in the latent space are actually close to the geodesic paths, given that we will not even know if the latent space characterization is accurate?

2. Another important issue in my opinion is the nature of "network inaccuracies". The paper only considers missing links. In practice however we can have not only false negative but also false positives (spurious links). A more realistic "error model" may actually show that the main result of the paper does not hold: namely that the shortest paths in latent space are not a very good approximation of geodesic paths even if we also consider false positives.

3. I would argue that in many applications, including the Internet and biological applications the paper considers, we do not want just to estimate the length of the shortest path between two nodes. We need to know the actual shortest path (or if there are several, we need to know ALL shortest paths because any of them could be chosen). From this point of view, I do not think that the paper shows strong results that the proposed approach can find actual shortest paths, or all of the shortest paths, reasonably well. So, the title of the paper ("Finding shortest and nearly shortest path nodes in large substantially incomplete networks") may be misleading.

Reviewer #2 (Remarks to the Author):

The authors applied hyperbolic embedding to find nodes in shortest and nearest shortest paths. They found that when the networks are largely incomplete, the hyperbolic embedding method is remarkably superior to the network-based method. The overall quality of this submission is high, but the title and organization of this submission is misleading. Specific comments and suggestions are as follows.

[1] The critical finding is this work is indeed not essentially related to the identification of the so-called shortest path nodes, but the higher reliability to link removal of hyperbolic-based metrics, compared with routine topological metrics. The current title and organization, although attractive and relevant to real-world applications, cannot well reflect the main finding. Therefore, the authors should change the title and reorganize the manuscript, and implement more experiments to show the advantage in using hyperbolic embedding, facing to missing data.

[2] The authors should clarify the network-based method (i.e., what does $l(A,B)$ mean in d_{nb}), is it just the topological distance?

[3] To be fair and to validate the superiority, the authors should compare their method with some other network-based distance metrics that may be more reliable to missing data, such as the random-walk-based distances [Fouss F, Pirotte A, Renders JM, Saerens M. Random-walk computation of similarities between nodes of a graph with application to collaborative recommendation, IEEE Transactions on Knowledge and Data Engineering 19 (2007) 355-369].

[4] In addition to [3], to show the superiority of the hyperbolic embedding method, the authors should also compare their method with other network embedding techniques (see Ref. 7 and the Section "Network Embedding" in [Zhou T. Progresses and challenges in link prediction, iScience 24 (2021) 103217]), such as node2vec, DeepWalk and GraphGAN.

[5] Real networks using in the experiments are too few.

[6] No caption for figure 1d.

Reviewer #3 (Remarks to the Author):

This paper conducts a fundamental study on finding the shortest and nearly shortest path nodes in large incomplete networks. The study mainly aims to find alternative shortest nodes for a target path, which has several potential important applications, such as cellular pathway reconstruction and communication security problems. The main finding of this paper is that shortest paths in large real networks are not random but are organized with latent-geometric rules. Based on this, the authors propose to map nodes in the original network space into latent hyperbolic spaces, so that the organized properties could be revealed in hyperbolic space. The findings of the main text are very interesting and convincing, which could be fundamental support for further studies. And the presentation, especially the figures in the paper, is very clear. I would like to list the strong points below.

Strong points:

1. The paper conducts fundamental research on nearly shortest path nodes found in large incomplete networks. The motivation that most real-world network is incomplete is convincing to me, as we could only observe partially data from the real latent space. Hence, the proposed method in hyperbolic space aims to robust with such missing links.
2. The finding results are very interesting, which clearly show that embedding nodes into hyperbolic space could help us reveal latent geometric rules. And the accuracy of the proposed method is much better than the results of the network-based method.

3. The paper also illustrates several very important applications of nearly shortest path nodes findings in the real world. For example, distance to geodesic can be invaluable in the validation of communication paths of a distributed communication network and finding the alternative nearly shortest paths of both communication networks and cellular pathways. I also would like to list several unclear points of the current version, hoping the authors could provide an explanation in the revision.

1. Given a network, how to judge whether it is organized with latent-geometric rules or not.

2. In 117 line page 6, network-based metric $d_{\text{nb}}(C|I_{\{A,C\}} + I_{\{A,B\}})$ is the sum of the network-based distances from node C to path endpoints A and B. To my understanding, it should be $d_{\text{nb}}(C|I_{\{A,C\}} + I_{\{C,B\}})$. If not, why is it defined as such formulation?

3. What's the meaning of the y-axis of Figure 3d?

4. Typos:

[line 108 page 6] "larger then" -> "larger than"

[line 312 page 16], $\text{Prob}(\{r_i, \theta_i\})$

We would like to thank the referees for reviewing our work and providing invaluable comments and constructive criticism, which helped us to greatly improve the clarity and the contents of our manuscript.

The major improvements of our work include:

- 1) The enhanced discussions of our findings and their applications,
- 2) The extension of our work to spurious (false positive) links,
- 3) Alternative random-walk-based methods for path finding accuracy, and a third real-world network.

We provide detailed responses to reviewers' comments below. For convenience, we provide a "marked-up" version of the revised manuscript where we highlight all changes in red.

REVIEWER COMMENTS

Reviewer #1 (Remarks to the Author):

The main point of this work is that if we map the nodes of a network into a latent geometric (hyperbolic to be precise) space, then the shortest paths in that space are often reasonably close (but not exactly -- and sometimes with substantial errors) to the shortest paths in the "native" network. Additionally, the paper makes the case that if we do not know the network accurately, in the sense that many real links may be missing from our inferred network, we can still estimate "almost shortest paths" reasonably well. The paper presents some results and applications of this approach on an Internet AS-level topology and on a similarity-based protein-protein interaction (PPI) network for Homo Sapiens.

The paper is well written, clear in its goal and methodology, and quite thorough and methodic in its scientific approach.

On the negative side, I do not think it represents a significant contribution -- something that could have either practical impact or that it adds important new knowledge in the study of complex networks. I will explain the reasons for this negative assessment next:

We thank the Reviewer for the pertinent summary of our work, and we are pleased that the Reviewer found our work "*quite thorough and methodic in its scientific approach*". At the same time, we feel that the manuscript, in its original form, lacked certain clarity which led to the perceived lack of significance by the first Reviewer. We hope that the clarifications provided in the manuscript and this document will alleviate this perception.

1. As the paper briefly admits in the Discussion, this entire approach is based on the assumption that the network has been formed in a way that is consistent with

a well-characterized geometric embedding. The authors correctly write: "one cannot expect to identify shortest paths in non-geometric networks using geometric methods." I would argue that it is actually more challenging than that: we need to know well the underlying geometry of the network.

Now the real issue here is that if the network is inaccurately known, which is the main "application focus" of this work, we may not be able to characterize that underlying geometry -- simply because we do not know about many links. If the network is inaccurately inferred, then even if we think that it is consistent, say, with hyperbolic embeddings, in reality it may not do so.

So how can we be sure that the paths we compute in the latent space are actually close to the geodesic paths, given that we will not even know if the latent space characterization is accurate?

The Reviewer asks a very deep question, which we try to address from several angles.

1. The reviewer is absolutely correct that our work is based on the assumption that the network of interest is intrinsically geometric. ***Can we assess the geometricity of a network prior to its embedding?*** We are afraid that despite the decades of research on network embeddings, statistical inference and machine learning, the community lacks mathematically founded criteria allowing us to **directly** assess network's geometricity.

At the same time, there are many **indirect** topological signatures of network geometricity, including sparsity, high density of short loops, and self-similarity (Ref. 8 in the revised manuscript). The three networks studied in our work have been assessed for their geometricity in prior studies and were shown to have properties consistent with intrinsic geometry (see Ref. 34 for the Internet, Ref. 9 for the PPI network, and Ref. 11 for the PGP web of trust).

We have added this important issue to the Discussion, lines 298-320 in the main text.

2. **Why do we expect the hyperbolic network map to be robust to missing links in the first place?**

Intuitively, geometric embedding can be viewed as a "mean-field image" of a network. In our approach, node coordinates are determined by maximizing the likelihood that the network of interest is generated in hyperbolic space with given specified network coordinates, see Methods and Section SIV. As a result, optimal (or equilibrium) coordinates of every node are determined by those of its neighbor nodes (that effectively attract the node) and all non-neighbor nodes (that effectively repel the node). If the network links are removed uniformly at random, intuitively, one

will expect that the effective forces are still balanced, and the “equilibrium” does not shift far from its location in the original complete network. Thus, latent-geometric geodesics and distances from nodes to them would not be significantly affected. This is not the case with network-based path-finding methods: even a single missing or spurious link may completely change the shortest path.

We revised our manuscript to better stress this aspect, both in the main text (lines 159-161) and Supplementary Section SIVA.

3. *How can we be sure that the paths we compute in the latent space are actually close to the geodesic paths, given that we will not even know if the latent space characterization is accurate?*

Strictly speaking, If our starting point is that we do not know the network accurately and, a priori, everything that we infer about its geometry is inaccurate, then we, indeed, should not even try to find shortest path through geometric embedding. By the same logic, however, we should not try finding shortest path with conventional network-based methods either.

From a more permissive standpoint, while we do expect inaccuracies in our path-finding results, we believe that the accuracy of the latent-geometric approach is higher than that of the network-based alternatives.

To verify this claim, we conducted a series of path-finding experiments on incomplete networks of variable degrees of completeness. In our experiments, we aim to identify nodes of nearly shortest paths in **original** networks (both real networks and synthetic models with known true coordinates) by learning node coordinates of their **incomplete** versions. Our results, depicted in Figs.~2, S8-S15 indicate that even though network topology may be inaccurately known (due to missing links) one can learn network coordinates with sufficiently high accuracy to identify shortest path nodes.

Furthermore, to better answer the Reviewer’s question we asked *how do the embedding coordinate inaccuracies translate into that of path-finding?* To answer this question, we studied the tolerance of latent-geometric path-finding to node coordinate uncertainties, which we modeled as synthetic noise added to learned node coordinates. Our results, shown in Fig. 5, indicate that our maps of the incomplete Internet network could tolerate additional coordinate uncertainty.

Summary of manuscript changes:

We fully rewrote the Discussion section to properly discuss conditions for latent-geometric path-finding. We added Fig. 5 to present the noise tolerance results. We extended Section SIV to discuss why hyperbolic network embeddings are expected to be robust to uniformly missing links.

2. Another important issue in my opinion is the nature of "network inaccuracies". The paper only considers missing links. In practice however we can have not only false negative but also false positives (spurious links). A more realistic "error model" may actually show that the main result of the paper does not hold: namely that the shortest paths in latest space are not a very good approximation of geodesic paths even if we also consider false positives.

We thank the Reviewer for this excellent suggestion to consider spurious links alongside missing ones to address both types of errors in the network data.

Indeed, spurious links are also common in real networks. While our work focuses on missing links and not on spurious links, we do agree that the former often occur in the presence of the latter. Therefore, we decided to stress-test our latent-geometric path-finding approach by also considering spurious links.

To this end, we have conducted two sets of experiments.

First, we added to real networks 10% (compared to the original number of links) of spurious links uniformly at random and repeated all missing link experiments. Our results depicted in Fig.~S14 remain qualitatively the same, indicating that latent-geometric path-finding methods remain applicable when a small fraction of spurious links are present.

To test the effects of a large fraction of spurious links present, we examined pathfinding in the setting of only spurious links connecting network nodes uniformly at random, varying the rate of spurious links between 30% and 90%. Our results indicate that the accuracy of our latent-geometric path-finding method remains high, Fig. S15. However, network-based path-finding methods are less sensitive to spurious links, eliminating the competitive advantage of the latent geometric method for the Internet and the similarity PPI networks.

As we explain in the revised Section SVIII, this is the case since randomly added spurious links tend to connect small degree nodes, while shortest and nearly shortest paths tend to traverse mid and high-degree nodes. Therefore, a spurious link is likely to affect a shortest path of interest only if it connects nodes in the vicinity of the both path endpoints simultaneously.

Summary of manuscript changes:

We added new figures S14 and S15 to present our path-finding experiments, and extended Section SVIII (SVII in the original manuscript) to discuss spurious links. We also discussed spurious links in the main text, lines 166-176.

3. I would argue that in many applications, including the Internet and biological applications the paper considers, we do not want just to estimate the length of the shortest path between two nodes. We need to know the actual shortest path (or if there are several, we need to know ALL shortest paths because any of them could be chosen). From this point of view, I do not think that the paper shows strong results that the proposed approach can find actual shortest paths, or all of the shortest paths, reasonably well.

We wholeheartedly agree with the reviewer here: Indeed, we need to know the paths themselves rather than the lengths of these paths. Furthermore, we fully agree that we need to know ALL feasible paths connecting the endpoints of interest.

As a matter of fact, we would like to emphasize that this is precisely what our work does.

The reviewer's perception is likely due to the lack of clarity in the original text, which we corrected in the revision.

Our work finds *nodes* constituting shortest *and* nearly shortest paths. Indeed, the proposed distance to geodesic allows for quantifying proximities of all network nodes to the geodesic: the closer the node to the geodesic the higher the likelihood that the node either belongs to the original shortest paths or may become shortest if the network topology is perturbed. In other words, distance to geodesic quantifies the node's relevance to **all** possible shortest paths. It is the ability of distance to geodesic to find **all** relevant paths in a network even if a large fraction of network links is missing that constitutes the impact of our work.

Also, it is the ability of distance to geodesic to determine nodes of all feasible paths that enables the proposed applications to Internet routing and cellular pathways.

Summary of manuscript changes:

After critically re-reading our manuscript, we realized that we do not adequately spell out the practical aspect of our findings until the middle of the manuscript. To better emphasize the practical value of our work and to put the results into the right perspective, we, therefore, updated the main text accordingly, lines 48-51, and 326-334.

So, the title of the paper ("Finding shortest and nearly shortest path nodes in large substantially incomplete networks") may be misleading.

We believe that the reason the Reviewer finds the title misleading follows from the perceived contribution issue, which we discussed above. As we explain in the response above, our approach allows one to find path nodes and not the lengths of the paths.

Therefore, we hope that addressing the above issue in the revised version of the manuscript will uphold the existing title.

At the same time, we believe it is of utmost importance to have a title correctly reflecting the content and the conclusions of our work. Therefore, we provide alternative titles for potential consideration, and we are more than happy to work out other alternatives in consultation with the editor and the Reviewers.

First, we would like to explain its original title

- a) In our work, we aim to find not only current shortest paths but also other feasible (or nearly shortest) paths that may be used if the original ones are inaccessible. This is the reason the original title contains "***Finding shortest and nearly shortest path nodes***".
- b) Distance to geodesic allows one to find **nodes** constituting shortest and nearly shortest paths: the smaller the distance to geodesic the higher the likelihood the node belongs to a path.

At the same time, it is important to emphasize what our approach *cannot* achieve. First, distance to geodesic cannot determine the network-based length of a path. Second, to define a path, one needs to specify not only its nodes but also their order in it. Since the distance to geodesic alone cannot determine the order of nodes in a path, we only find "path nodes", not paths *per se*.

This is the reason, the original title says "***Finding shortest and nearly shortest path nodes***".

- c) We emphasize **large** networks for two reasons: (i) hyperbolic network embedding is primarily designed for large networks, (ii) path-finding is badly needed in large incomplete networks where one cannot search for paths and/or missing links exhaustively.

Since our work proposes to find paths by embedding them into latent space, and we focus on hyperbolic latent spaces, we propose to modify the existing title to

V1: "Finding shortest and nearly shortest path nodes in large substantially incomplete networks by hyperbolic mapping".

Other alternative options we can offer are:

V2: "Finding shortest paths in large networks with missing and spurious links"

or

V3: “Hyperbolic mapping enables finding shortest paths in large networks with missing and spurious links”

Summary of manuscript changes:

We have updated the title using the first version.

Reviewer #2 (Remarks to the Author):

The authors applied hyperbolic embedding to find nodes in shortest and nearest shortest paths. They found that when the networks are largely incomplete, the hyperbolic embedding method is remarkably superior to the network-based method. The overall quality of this submission is high, but the title and organization of this submission is misleading. Specific comments and suggestions are as follows.

[1] The critical finding is this work is indeed not essentially related to the identification of the so-called shortest path nodes, but the higher reliability to link removal of hyperbolic-based metrics, compared with routine topological metrics. The current title and organization, although attractive and relevant to real-world applications, cannot well reflect the main finding. Therefore, the authors should change the title and reorganize the manuscript, and implement more experiments to show the advantage in using hyperbolic embedding, facing to missing data.

We are glad that the Reviewer found the overall quality of our submission to be high. At the same time, we feel that the Reviewer finds our title misleading for the same reason the first Reviewer did - the lack of the details on our side to properly highlight manuscript contributions.

In short, the main practical result of our work is that the observed geometric localization of shortest paths can be used to identify shortest path nodes. To do so, we propose distance to geodesic: the closer the node to the geodesic the higher the likelihood the node belongs to the shortest path.

We would like to refer the second Reviewer to our extended discussion of the issue in our response to the third comment of the first Reviewer, where we justify the original title and propose three alternative versions.

We have also followed the Reviewer’s suggestions and implemented additional experiments in our work, which we discuss below. We hope that the Reviewer finds the revised manuscript to be forthright and not misleading.

[2] The authors should clarify the network-based method (i.e., what does $l(A,B)$ mean in d_{nb}), is it just the topological distance?

It is indeed just a topological distance or a network-based length of the shortest path connecting A and B.

Summary of manuscript changes:

We have clarified d_{nb} in the main text, lines 122 and 353-355.

[3] To be fair and to validate the superiority, the authors should compare their method with some other network-based distance metrics that may be more reliable to missing data, such as the random-walk-based distances [Fouss F, Pirotte A, Renders JM, Saerens M. Random-walk computation of similarities between nodes of a graph with application to collaborative recommendation, IEEE Transactions on Knowledge and Data Engineering 19 (2007) 355-369].

We thank reviewer for this excellent suggestion, which allowed us to greatly enhance the scope of the path-finding assessment. Following this suggestion, we have implemented two random-walk-based metrics and evaluated their accuracies in finding shortest paths in all networks under all conditions. Moreover, we have added an entirely new network for our analysis, per the Reviewer's suggestions to add more networks. The results are reflected in both the main and supplementary text and also in Figures. 2, S12, S13, S14, and S15.

Following the Reviewer's suggestion, we compared the accuracy of distance to geodesic to that of the metric based on the average commute time $n(X,Y)$. In more precise terms, we have quantified the deviation of node C from nearly shortest path A-B as $n(A,C) + n(C,B)$. Following the proposed reference, we computed the average commute times through the pseudoinverse of the Laplacian. We were able to compute the pseudoinverse matrices for the PPI network and the PGP network web of trust (the latter is a new network we added to our analysis).

Due to the large size, $N > 23,000$, we were not able to compute average commute distances for the Internet network reliably. (Which indicates the limited applicability of the pseudoinverse of the Laplacian metric to large networks)

Therefore, we introduced another measure based on directly computing random walks, which we called the random walk hit frequency, see Methods, and used it in all experiments. Confirming the Reviewer's remark, we found that the random-walk-based metrics are less sensitive to missing data compared to the network-based metric. Nevertheless, we found that the overall accuracies of random-walk-based measures to be subpar than those of the distance to geodesic.

Summary of manuscript changes:

We have updated the path-finding performance figures, Fig. 2, and Figs. S12 - S15, with the random-walk-based results. We have expanded the discussion of path-finding methods, and extended Section SVIII, Methods, and the main text, lines 162-165.

[4] In addition to [3], to show the superiority of the hyperbolic embedding method, the authors should also compare their method with other network embedding techniques (see Ref. 7 and the Section “Network Embedding” in [Zhou T. Progresses and challenges in link prediction, *iScience* 24 (2021) 103217]), such as node2vec, DeepWalk and GraphGAN.

We appreciate the Reviewer’s suggestion to compare the hyperbolic embedding to other popular embedding methods. In the revised manuscript, we have studied the performance of the node2vec and the DeepWalk embedding methods. Our findings imply that high-dimensional *Euclidean* embedding methods are not sufficiently accurate for shortest path-finding tasks, in their present form. Intuitively, Euclidean spaces do not conform to the hierarchical organization of many large networks. The number of neighbors reachable from a node of interest grows exponentially with the number of link hops away from the node. At the same time, the volume of Euclidean space is confined with distance R from a point of interest (the volume of a Ball) grows polynomially. As a result, we do not expect shortest paths in real networks to follow *Euclidean* geodesics.

Summary of manuscript changes:

We have added a discussion of alternative embedding methods in the main text, lines 299-312, and added Fig. 5 to support it.

[5] Real networks using in the experiments are too few.

Indeed, the original manuscript analyzes only two large real-world networks. Our goal in this work is not only to evaluate the accuracy of hyperbolic path-finding method and its robustness to missing network data, but also to identify principal domain-relevant applications where this method will make an immediate impact. These applications are the validation of routing paths and the analysis of cellular pathways. The in-depth of these applications on the large scale is not only time-consuming but also requiring domain-level expertise. This is the reason why in the original version we focused only on two real networks.

The two considered networks – the Internet and the similarity PPI network – are examples from technological and biological domains. Another domain where finding paths is needed is the domain of social networks. Therefore, following the Reviewer’s suggestion, we added another (social) network to our consideration. The network is the Pretty-Good-Privacy (PGP) web of trust. **We replicated all the analyses done for the first two networks.** Our path-finding results on the PGP web of trust agree with those obtained for the Internet and the similarity-based PPI network.

Summary of manuscript changes:

Additional Section SVII is dedicated to the description of the PGP web of trust. Figures S13 and parts of Figs. S14 and S15 are dedicated to path-finding experiments on the PGP web of trust.

[6] No caption for figure 1d.

We apologize for the omission.

Summary of manuscript changes:

We have added the caption for Fig. 1d in the revised manuscript.

Reviewer #3 (Remarks to the Author):

This paper conducts a fundamental study on finding the shortest and nearly shortest path nodes in large incomplete networks. The study mainly aims to find alternative shortest nodes for a target path, which has several potential important applications, such as cellular pathway reconstruction and communication security problems. The main finding of this paper is that shortest paths in large real networks are not random but are organized with latent-geometric rules. Based on this, the authors propose to map nodes in the original network space into latent hyperbolic spaces, so that the organized properties could be revealed in hyperbolic space. The findings of the main text are very interesting and convincing, which could be fundamental support for further studies. And the presentation, especially the figures in the paper, is very clear. I would like to list the strong points below.

Strong points:

- 1. The paper conducts fundamental research on nearly shortest path nodes found in large incomplete networks. The motivation that most real-world network is incomplete is convincing to me, as we could only observe partially data from the real latent space. Hence, the proposed method in hyperbolic space aims to robust with such missing links.*
- 2. The finding results are very interesting, which clearly show that embedding nodes into hyperbolic space could help us reveal latent geometric rules. And the accuracy of the proposed method is much better than the results of the network-based method.*
- 3. The paper also illustrates several very important applications of nearly shortest path nodes findings in the real world. For example, distance to geodesic can be invaluable in the validation of communication paths of a distributed communication network and finding the alternative nearly shortest paths of both communication networks and cellular pathways.*

We thank the Reviewer for the pertinent summary of our work and for highlighting its strong points. We are pleased that the Reviewer finds the presentation to be clear and that “*the findings of the main text are very interesting and convincing*”.

I also would like to list several unclear points of the current version, hoping the authors could provide an explanation in the revision.

1. Given a network, how to judge whether it is organized with latent-geometric rules or not.

This is an excellent question, which the community does not have a full answer for, unfortunately.

Despite the decades of research on network embeddings and other geometric methods, we still lack the direct assessment tools for network geometricity. One criterium that seems to hold for all geometric networks is a high density of short loops in a network, which can be quantified e.g., with clustering coefficient. One reference, to this end, is “Clustering implies geometry in networks”, *Phys. Rev. Lett.* 116, 208302 (2016), argues that large numbers of triangles, homogeneously distributed across all nodes as in real networks, are thus a consequence of network geometricity. All networks studied in our work, both real and synthetic are characterized by a strong clustering coefficient, in agreement with this work.

For a summary of results pertaining to network geometricity, we can recommend a review paper *Network Geometry*, in *Nature Review Physics*, 3 (2), 114-135 (2021) by M. Boguñá et al.

Summary of manuscript changes:

We have extended the Discussion section to address this question in the main text.

2. In 117 line page6, network-based metric $d_{\text{nb}}(C||_{\{A,C\}} + l_{\{A,B\}})$ is the sum of the network-based distances from node C to path endpoints A and B. To my understanding, it should be $d_{\text{nb}}(C||_{\{A,C\}} + l_{\{C,B\}})$. If not, why is it defined as such formulation?

Summary of manuscript changes:

The Reviewer is correct, and we have corrected the typo accordingly.

3. What's the meaning of the y-axis of Figure 3d?

The y-axis of Fig. 3d is the probability density, and the plot depicts the pdf that a randomly selected BGP update has a specified stretch.

Summary of manuscript changes:

We have added a title to the y-axis in Fig. 3d.

4. Typos:

[line 108 page 6] “larger then” -> “larger than”
[line 312 page 16], $Prob(\{r_i, \theta_i\})$

We thank the Reviewer for finding these typos. We have corrected them in the revised manuscript.

REVIEWERS' COMMENTS

Reviewer #1 (Remarks to the Author):

First, I would like to recognize that the authors have done an excellent job in responding to the reviews and in revising their paper. The responses are clear, relevant, and honest -- accepting some challenges and limitations of the study without trying to "oversell" what the paper does. The new results (mostly about the performance of the method in the presence of missing or spurious links) are important and they add value in the paper.

Referring to my original review, my main concern was related to the claimed significance of this study -- and I am afraid that my opinion has not changed much after the revisions. The authors acknowledge that, strictly speaking, 1) we cannot be certain about the "geometricity" of a network when many links are missing, 2) the proposed method does not identify shortest paths but nodes that are close to shortest paths. Of course they also provide arguments to reduce the seriousness of these limitations -- but whether those arguments are convincing is rather subjective.

So to summarize my opinion about this paper:

on the positive side, it is certainly a solid piece of research and a contribution in the literature of geometric embeddings in complex networks.

on the negative side, I think it is an incremental contribution -- something like an "expected next step" in a long sequence of previous research contributions.

Whether this is "above the bar" for the Nature Communications journal, I will leave it up to the Editor to decide, also considering the opinion of the other reviewers.

Reviewer #2 (Remarks to the Author):

The authors have well addressed issues in the last report.

Reviewer #3 (Remarks to the Author):

After the revision, although the authors could not give a doubtless conclusion for my first concern, I still feel that learning the shortest path from the latent-geometric space is an interesting idea. And the paper provides sound experimental results for their findings. From my aspect, the authors have addressed my concerns. And I also notice that other reviewers have some important concerns that I agree with. If the authors could also address these concerns, the paper could be accepted.

We would like to thank the referees for reviewing our work and providing invaluable comments and constructive criticism.

Reviewer #1 (Remarks to the Author):

First, I would like to recognize that the authors have done an excellent job in responding to the reviews and in revising their paper. The responses are clear, relevant, and honest -- accepting some challenges and limitations of the study without trying to "oversell" what the paper does. The new results (mostly about the performance of the method in the presence of missing or spurious links) are important and they add value in the paper.

We are glad that the reviewer finds our responses clear, relevant, and honest.

Referring to my original review, my main concern was related to the claimed significance of this study -- and I am afraid that my opinion has not changed much after the revisions. The authors acknowledge that, strictly speaking, 1) we cannot be certain about the "geometricity" of a network when many links are missing,

The reviewer is right, we lack the direct methods to assess the geometricity of a network. The problem, however, is not specific to incomplete networks. Even if the network topology is fully known, we lack direct methods that would assess the "likelihood" the network is geometric. At the same time, there are multiple works dedicated to "indirect" geometricity signatures. To this end, the topological properties of spatial network models, are relatively well studied. These properties include sparsity, strong clustering coefficient, and self-similarity. Many real networks, including the Internet, the PPI network, and the PGP web of trust studied in our work, have been shown to possess these properties.

To better highlight the hardness of the network geometricity problem, we have included these arguments in the Discussion section (lines 310 – 319 in the revised manuscript).

2) the proposed method does not identify shortest paths but nodes that are close to shortest paths. Of course they also provide arguments to reduce the seriousness of these limitations -- but whether those arguments are convincing is rather subjective.

We observe that nodes with the smallest distance to geodesic are both shortest path nodes and nearly shortest path nodes, as defined by the node's path relevance, Fig. 1d. We discuss this aspect in Methods, lines 355-361, and Results, lines 109-117.

At the same time, we would like to acknowledge that our study does bring new research questions to the table. While our results clearly indicate that distance to geodesic can

find nearly shortest path nodes better than alternative strategies if the network is incomplete, we observe that precision of distance to geodesic is well below 100% for incomplete networks. This motivates the following questions: “Which shortest path nodes are left behind by distance to geodesic?” “Are there “hard-to-find” shortest path nodes?” “Can distance to geodesic be combined with another path-finding approach to better cover the “hard-to-find” nodes? We hope to address these questions in follow-up studies.

So to summarize my opinion about this paper:

on the positive side, it is certainly a solid piece of research and a contribution in the literature of geometric embeddings in complex networks.

on the negative side, I think it is an incremental contribution -- something like an "expected next step" in a long sequence of previous research contributions.

Whether this is "above the bar" for the Nature Communications journal, I will leave it up to the Editor to decide, also considering the opinion of the other reviewers.

We do believe that our finding of geometric localization of shortest and nearly shortest path nodes constitutes a significant contribution since it allows to identify these nodes in substantially incomplete networks. However, we do agree with the reviewer that the ultimate decision if our work is sufficiently significant for *Nature Communications* is up to the Editor to decide.

Reviewer #2 (Remarks to the Author):

The authors have well addressed issues in the last report.

We would like to take another chance to thank the reviewer for the insightful comments and for reviewing our responses.

Reviewer #3 (Remarks to the Author):

After the revision, although the authors could not give a doubtless conclusion for my first concern, I still feel that learning the shortest path from the latent-geometric space is an interesting idea. And the paper provides sound experimental results for their findings. From my aspect, the authors have addressed my concerns. And I also notice that other reviewers have some important concerns that I agree with. If the authors could also address these concerns, the paper could be accepted.

The reviewer's first concern is "*How to judge whether it is organized with latent-geometric rules or not?*" This concern is also related to that of the first reviewer's comment about one's ability to assess the geometricity of an incomplete network.

The above question is an important open problem for which the community does not have a direct answer. While we lack the knowledge of "sufficient" conditions for network geometricity, we do have knowledge of the "necessary" conditions. For instance, it has been established that geometric network models are characterized by sparsity, strong clustering coefficient, and self-similarity, and many real networks also possess these topological properties. One possible way to prove that these necessary conditions are also sufficient is to establish maximally random network ensembles subject to the above topological constraints (sparsity, clustering, and self-similarity) are equivalent to spatial network ensembles. The first step, to this end, has been done in Ref. [35] (of the revised manuscript) by establishing the equivalence between network ensembles with fixed clustering and expected degree and random geometric graphs on a straight line.

To better highlight the hardness of the network geometricity problem, we have included these arguments in the Discussion section (lines 310 – 319 in the revised manuscript).